# Efficient Feature Learning Approach for Raw Industrial Vibration Data Using Two-Stage Learning Framework

**DOI:** 10.3390/s22134813

**Published:** 2022-06-25

**Authors:** Mohamed-Ali Tnani, Paul Subarnaduti, Klaus Diepold

**Affiliations:** 1Department of Factory of the Future, Bosch Rexroth AG, Lise-Meitner-Str. 4, 89081 Ulm, Germany; 2Department of Electrical and Computer Engineering, Technical University of Munich, Arcisstr. 21, 80333 Munich, Germany; ge73gid@mytum.de (P.S.); kldi@tum.de (K.D.)

**Keywords:** feature learning, CNC machining, machine monitoring, machine learning, few-shot learning, vibration data, two-stage learning

## Abstract

In the last decades, data-driven methods have gained great popularity in the industry, supported by state-of-the-art advancements in machine learning. These methods require a large quantity of labeled data, which is difficult to obtain and mostly costly and challenging. To address these challenges, researchers have turned their attention to unsupervised and few-shot learning methods, which produced encouraging results, particularly in the areas of computer vision and natural language processing. With the lack of pretrained models, time series feature learning is still considered as an open area of research. This paper presents an efficient two-stage feature learning approach for anomaly detection in machine processes, based on a prototype few-shot learning technique that requires a limited number of labeled samples. The work is evaluated on a real-world scenario using the publicly available CNC Machining dataset. The proposed method outperforms the conventional prototypical network and the feature analysis shows a high generalization ability achieving an F1-score of 90.3%. The comparison with handcrafted features proves the robustness of the deep features and their invariance to data shifts across machines and time periods, which makes it a reliable method for sensory industrial applications.

## 1. Introduction

The latest advances in technology coupled with an aim to realize smart intelligent systems have contributed to a rapid move towards the next industrial revolution. Unlike the third industrial revolution powered by electronics and information technology, digitization and automation have been the front runners to revolutionize industry to its fourth chapter. The fourth industrial revolution has proved to be a boon to the traditional machining processes as it brings some key advantages such as improvement in the production and quality, cost reduction, and monitoring of machining processes in real time. As a result, condition monitoring and process condition monitoring systems are integral parts of intelligent manufacturing that support the quality inspection. Such highly automated systems rather support the flow of huge volumes of data that can be analyzed in real time without interrupting any machining workflow [1].

Enabled by the significant advancements in industrial Internet of Things (IIoT), the process involved in collecting and monitoring data from industrial environment is made more convenient. The initial step usually involves the acquisition of different types of signals such as vibration, cutting force, and a few others that can determine the health of machining parts and tool processes. This work largely focuses on the vibration-based signals as it provides critical information about the machining health. However, the vibration signals collected from the sensors are largely affected by several environmental factors and are commonly characterized by their nonlinearity, nonstationarity and noisiness. This brings us to the next steps of monitoring systems that are filtering the collected signals [2]. Feature extraction as means of signal filtering is a crucial step in the data processing pipeline. With the gradual development in machine learning (ML) algorithms and eventually deep neural networks, the idea of feature extraction from the raw vibration signals has varied over time. Traditionally, the feature extraction mainly involved signal processing techniques such as statistical analysis on the time, frequency or time–frequency domain [3,4,5,6]. Although these techniques have produced fair results over the years, they also present some major drawbacks. These algorithms often require extensive domain knowledge as well as human expertise specifically designed for a specified task. As the volume of the collected data increases, which is particularly huge in modern automated smart systems, the effort as well as time to produce meaningful representation increases. One implementation done by Christ et al. named as TSFRESH [7] has achieved remarkable results. It can automatically extract statistically based features and observe dynamics without much human expertise. Lines et al. [8] also presented a hierarchical transformation ensemble method for time series classification. However, these methods fail to meet the demands of a fast reliable algorithm due to the high computational time.

Recently, encouraged by the outstanding performance of deep learning (DL) in several fields, some interesting end-to-end DL algorithms have been proposed to replace traditional time-consuming monitoring systems. Unlike image processing, research in machine monitoring has mostly overlooked the advantages of deep neural networks due to their hard interpretability and their nonacceptance in industry [9]. Nevertheless, state-of-the-art research works [10,11] have integrated DL techniques on vibration data that are treated as one-dimensional time series data and showed state-of-the-art results, bypassing the handcrafted-based methods. However, these supervised methods require a huge quantity of labeled data to achieve satisfying performance. Data annotation is another critical factor in real-world production plants, as labeling large quantities of data is often an inconvenient, costly, and erroneous approach under human supervision. Moreover, in a highly automated system, the occurrence of anomalies is a rare event that causes a huge imbalance between OK and NOK samples. These factors deteriorate the performance of supervised DL algorithms that fail to generalize on noisy time-series (TS) data. To tackle these shortcomings, unsupervised feature extraction technique has proved to be promising. In particular, autoencoders have been found to be most beneficial algorithm [12]. Sun et al. [13] showed that a sparse autoencoder with a small number of trainable parameters can learn good features based on induction motor data. Shao et al. [14] also proposed a work that addresses the generalization of autoencoders on unseen working conditions in fault diagnosis.

Despite its huge success, conventional DL techniques require huge quantities of data to offer meaningful generalization on unseen data. In the literature, the problem of insufficient labeled data samples has been handled in different ways. Data augmentation plays a crucial role in processing such raw vibration data. Overlapping input data samples to generate small snippets of new samples is one such technique used by [10]. In the fault diagnosis applications, a few works have showed how data augmentation could generate new synthetic samples using GANs [15,16]. However, they suffer from overfitting problems. To overcome the challenge of limited labeled data, certain ML algorithms named few-shot learning (FSL) methods have been proposed in the state-of-the-art literature [17]. Such a learning paradigm has been designed to tackle scenarios where data with appropriate labels are difficult to produce, such as in an industrial environment. Considering a small training dataset (x,y), FSL can be best described as an optimization algorithm that searches for the best hypothesis space from *x* to *y* described by the set of optimal parameters [18]. Current state-of-the-art literature has produced FSL for various applications mostly featuring computer vision tasks and only a few implementations can be found for time-series classification. The authors in [19,20,21,22] proposed a metalearning model for few-shot fault diagnosis applications. The prototypical network is also a popular FSL technique for time-series classification. It has proved to achieve state-of-the-art results for both few-shot and zero-shot classification problems [23,24]. Tang et al. [25] proposed a novel few-shot learning approach for time-series classification. In the feature learning research on rolling bearing fault diagnosis, Wang et al. [21] proposed a metric-based metalearning method named relational network which learns fault features from the input FFT frequency signals. A few studies [21,26] also explored few-shot learning for fault diagnosis on rotatory machines such as CNC machines.

To address the problem of costly data annotation and the imbalance between normal and abnormal machine faults, this work aims to propose a novel two-stage feature learning framework using the prototypical few-shot technique. Recently, researchers have largely benefited from the two-stage frameworks, which have gradually attracted a lot of attention. The existing methods in the state-of-the-art literature fall into two categories. The first category is the two-stage predicting category, which aims to improve the performance of the prediction task by decomposing the application task into two sequential tasks. Few studies [6,27,28] have explored the two-stage predicting category. To detect defective rolling element bearings, Yiakopoulos et al. [6] presented a two-stage method, where the first task was to detect the existence of a bearing fault while the second stage task classified the type of detected anomaly. The second category is the two-stage learning framework, which aims to improve the learning following a graduated training methodology. In the image processing field, Das et al. [29] tackled the problem of the high dimensionality and the variable variance among the base classes with a two-stage feature learning approach. The first stage produces a relative feature extractor, while the second stage handles the classification task by measuring the variance using distance metrics such as the Mahalanobis distance. Afrasiyabi et al. [30] aimed to represent rich and robust features from input images using mixture-based feature learning (MixtFSL). The proposed end-to-end approach learned in a progressive manner till the best feature representation was achieved. Ma et al. [31] proposed a two-stage training strategy called partner-assisted learning, where soft anchors were generated by a partner model in the first stage and the main encoder was trained by aligning its outputs with the soft anchors in the second stage. In wind turbine condition monitoring applications, Afrasiabi et al. [32] presented a sequential training pipeline that resolved the limited data problem by generating artificial data in the first stage and training a robust deep Gabor network in the second stage.

This work falls into the second category and proposes a novel two-stage feature learning framework for industrial machining processes. The study focuses on the performance of the resulting feature extractor trained with limited labeled data, its ability to generalize over unseen machining process operations with different working conditions as well as its robustness against data drift. The work is divided into sections. The second section presents the background of the prototypical network (PN) and the different distance measures used in this work. In the third section, we define the smart data sampling technique for noisy time series and the proposed two-stage learning approach. In the fourth section, we introduce the publicly available Bosch machining dataset and present a real-world scenario mapped in the dataset-splitting part. In the fifth section, we describe the experiments performed, followed by an in-depth analysis of the results as well as a comparison with different types of feature extractors. Finally, we conclude with a short summary and the prospect of some future work.

## 2. Background

### 2.1. Prototypical Networks

This work greatly focuses on prototypical networks [23] for few-shot learning. For an *N*-way and *K*-shot FSL, we have a small training dataset D with *k* labeled samples. $D=\{\left(x_{1},y_{1}\right),..,\left(x_{k},y_{k}\right)\}$, where $x_{i}$ represents a *D*-dimensional input feature vector and each $y_{i}$ represents its corresponding label. The training is divided into several episodes termed as training episodes. For each episode, training sets are sampled to form a support set S and a query set Q.
Support Set: A random subset of classes from the training set is selected as support set containing *K* examples from each of the *N* classes.Query Set: A set of “testing” examples called queries.

Taking each class into consideration, prototypical networks generate the embedded points for each example in S using an embedding function $f_{\theta }$. For each class $N_{k}$, a mean vector of the embedded points $C_{k}$ is computed using Equation (Equation 1) and represents the prototype of the $N_{k}$ class.
(1)$$\begin{array}{c} C_{k}=\frac{1}{|S_{k}|}\sum_{x\in S_{k}}f_{\theta }\left(x\right) \end{array}$$

By computing a distribution over classes, the prototypical network classifies the queries using a softmax function over the distances to the prototypes following Equation (Equation 2). Snell et al. [23] highlighted the significance of using a squared Euclidean distance as a distance function in image classification tasks. In this paper, we further studied different distancing functions for noisy time-series classification tasks.
(2)$$\begin{array}{c} P_{\theta }\left(y=k|x\right)=\frac{exp(-d\left(f_{\theta }\left(x\right),C_{k}\right))}{\sum_{k^{\prime }}exp\left(-d\left(f_{\theta }\left(x\right),C_{k^{\prime }}\right)\right)} \end{array}$$

Finally, the network is optimized by minimizing the negative log-probability of the true class with an Adam optimizer [33] and updating the parameters θ of the embedding function *f* using the loss Equation (Equation 3).
(3)$$\begin{array}{c} L\overset{}{\leftarrow }L+\frac{1}{N_{C}N_{Q}}\left[d\left(f_{\theta }\left(x\right),C_{k}\right)+log\sum_{k^{\prime }}exp(-d\left(f_{\theta }\left(x\right),C_{k}\right)\right] \end{array}$$

### 2.2. Distance Metrics

L2 Euclidean: Given two vector points *U*: ($u_{1}$,…, $u_{k}$) and *V*: ($v_{1}$,…,$v_{k}$), the Euclidean L2 distance is defined as the shortest distance between two vector points, a commonly used similarity metric in various applications.
(4)$$\begin{array}{c} D\left(U,V\right)=\sqrt{\sum_{i=1}^{k}{\left(u_{i}-v_{i}\right)}^{2}} \end{array}$$

DTW distance: DTW or dynamic time warping [34] was coined as a distance metric to find the similarities between two time sequences. Unlike the Euclidean distance, which is prone to both global and local shifts in time dimension, DTW tackles such unintuitive results and aims at finding the minimum warp path between two time sequences. Given two time sequences *P* and *Q* and their individual lengths |P| and |Q|, respectively, DTW constructs a warp path which is given by
(5)$$\begin{array}{c} W=w_{1},w_{2},..,w_{k},\;\mathrm{where}\;w_{k}=\left(i,j\right)\;\mathrm{and}\;w_{k}+1=\left(i^{*},j^{*}\right) \end{array}$$

The warp path begins at index (1,1) and ends at (|P|,|Q|). The optimal warp path Dist(W) is thereby given by the sum of the distances between the individual warp paths from index *i* in *P* to index *j* in *Q*, meaning
(6)$$\begin{array}{c} Dist\left(W\right)=\sum_{k=1}^{k=K}Dist\left(w_{ki},w_{kj}\right) \end{array}$$

To reduce the time complexity of DTW from $O(N^{2})$ to O(N), FastDTW has been proposed in the state-of-the-art literature [34]. Keeping the whole DTW algorithm similar, it introduces three constraints: coarsening (shrinking the time sequence into smaller time steps), projection (calculating the minimum warp distance at low resolution), and refinement (refining the low-resolution warp path through local adjustments) to reduce the time complexity.

Cosine distance: The cosine distance is another metric that is used to measure the similarity between two vector points. It measures the cosine of the angle between two vector points. The cosine similarity metric and cosine distance metric are correlated and can be found in the following equations.
(7)$$\begin{array}{c} Cosine\;similarity\left(U,V\right)=cos\left(\theta \right)=\frac{U.V}{∥U∥∥V∥}=\frac{\sum_{i=1}^{k}u_{i}v_{i}}{\sqrt{\sum_{i=1}^{k}u_{i}^{2}}\sqrt{\sum_{i=1}^{k}v_{i}^{2}}}\\ D\left(U,V\right)=1-Cosine\;similarity\left(U,V\right) \end{array}$$

## 3. Method

In this paper, we propose a generic feature learning method for monitoring machining processes using limited TS data annotations. In the next sections, the mixture-based data selection method is defined, followed by the proposed two-stage feature learning method.

### 3.1. Mixture-Based Data Selection

The learning performance depends mainly on the input data. This makes data selection not only the first but also a crucial step in FSL since it aims at choosing the training query and support sets. In computer vision, sample selection is straightforward given the standardized format of the image data. However, TS data, such as process vibration data, is characterized by the variation of signal length due to the different measurement lengths. This leads to an imbalance of data after data windowing and degrades the learning performance. In this work, we used a mixture-based data selection technique (MDS), which is illustrated in Figure 1.

The data signals x∈D in each class $N_{k}$ are first windowed using a sliding window with a fixed $w_{s}$. In online industrial applications, data are buffered in chunks, which explains the use of the sliding window when developing industrial data processing techniques and speeds up the computing by avoiding additional data analysis steps. The output of the first step is a set of same-shaped signals $x_{w}\in R^{w_{s}\;x\;C}$, where *C* is the number of channels. In the case of the vibration data used in this work, *C* was equal to 3 with reference to the {*X*, *Y*, *Z*} axes. For the sake of simplicity, only one axis of the vibration signal (*C* = 1) is shown in the MDS illustration in Figure 1. The windowing step is followed by a random selection step that samples the training sets, i.e., the query and support sets, during the episodic learning process. For an FSL task with *N* ways and *K* shots, the MDS outputs support set $S={\left(x,y\right)}^{N\;x\;K}$ and query set $Q={\left(x,y\right)}^{N_{Q}}$ with $N_{Q}$ being the number of queries per iteration. As stated above, the measurement length mismatch leads to an imbalance between the different subclasses, i.e., the different machining processes. The MDS sampling technique produces an equal number of data samples in the OK and NOK training sets at each training episode, which reduces the negative impact of the imbalance rate and results in more unbiased models. The second advantage of the MDS method is the high informativeness of the training sets in terms of the diversity of signals in each class. In fact, at each training episode, thanks to the windowing step followed by a random selection, the MDS leads to a more diverse selection of samples from different periods, machines, and processing operations, which allows the FSL models to be drift invariant and facilitates the search for discrepancy between the OK and NOK classes. This result increases the robustness of the feature extractor, which is insensitive to the challenging conditions in machining applications.

### 3.2. Two-Stage Learning Framework

The proposed method represents a two-stage learning framework for noisy industrial TS data and is shown in Figure 2. The first stage consists of an unsupervised pretraining stage, while the second step consists of fine-tuning the learned feature extractor using very limited annotations and is therefore referred to as the metalearning stage.

#### 3.2.1. First Stage: Unsupervised Pretraining

Industrial use-cases are characterized by their large volume of unlabeled data, in particular for time series data. In order to take advantage of the unannotated data and overcome the imbalance effect on supervised learning, the two-stage learning starts with an unsupervised feature learning using the autoencoder (AE) method [12]. In this phase, the encoder *f* with parameters θ learns the representation of the unlabeled dataset $D_{unlabeled}$ by encoding the input signal *x* into a compressed vector $x_{enc}$. The encoder architecture, which represents the deep feature extractor of the proposed method, was designed based on a convolutional neural network (CNN) and is illustrated in Figure 3. To best evaluate the two-stage learning method, a simple stacked CNN was chosen with 3 consecutive convolutional blocks followed by a final bottleneck layer. Each convolutional block consisted of a 1-D convolutional layer, a batch normalization layer [35], a ReLu (Rectified Linear Unit) activation function, and a max pooling layer.

The decoder $g_{\phi }$ is a transposition of the encoder $f_{\theta }$ and performs the decoding of the encoded feature vector $x_{enc}$ into the reconstructed signal $x_{rec}$. The objective function of the autoencoder *E* is the mean square error (MSE) between $x_{rec}$ and the input signal *x* according to the following equation:(8)$$\begin{array}{c} E=\frac{1}{n}\sum_{i=1}^{n}{\left[x-g_{\phi }\left(f_{\theta }\left(x\right)\right)\right]}^{2} \end{array}$$

The result of this phase consists of the pretrained parameters of the encoder function $f_{\theta_{pretrained}}$ and the decoder part is dropped. The training process of the first stage follows the pseudocode in Algorithm 1.
**Algorithm 1** First stage: unsupervised pretraining**Input:** Unlabeled data set $D_{unlabeled}$**Output:** Pretrained encoder function $f_{\theta_{pretrained}}$θ,ϕ← Initialize randomly**for** number of epochs **do**    compute MSE error *E* using Equation (Equation 8)    θ,ϕ← Update using gradients of *E*▹ compute backpropagation**end for**

#### 3.2.2. Second Stage: Metric Meta Learning Stage

The second stage consists of fine-tuning the unsupervised pretrained feature extractor $f_{\theta }$ for a specific task using very limited annotated dataset $D_{labeled}$ in an episodic manner. The first step consists of sampling the training steps using the Section 3.1 method resulting in highly informative support sets. For each signal in the support set, the embedded vector is extracted using the feature extractor $f_{\theta }$ and these deep feature vectors are then averaged by class. This results in *N* representative $C_{k}$ prototypes for each class. Using a distancing function, each prototype is then matched against each embedded query point, which is classified by simply finding the closest class prototype. The distancing function is crucial to the feature learning process as it defines the loss function *L* (Equation 3) and therefore the optimization of the feature extractor parameters. To find the optimal distance function for the noisy vibration data, we evaluated in the experimental section different TS measures (Euclidean, cosine, and DTW). The parameters θ are later updated using the gradients of the loss function *L* using the Adam optimizer function [33]. Once the metric metalearning stage is completed, the resulting feature extractor $f_{\theta }$ is evaluated on an unseen dataset $D_{test}$ and on the visualization of the embedding space of vibration data. The training process of the second stage follows the pseudocode in Algorithm 2.
**Algorithm 2** Second stage: metric-based fine-tuning**Input:** Labeled data set $D_{labeled}$, pretrained encoder function $f_{\theta_{pretrained}}$**Output:** Two-staged trained encoder function $f_{\theta }$$f_{\theta }\leftarrow f_{\theta_{pretrained}}$▹ initialize encoder with the pretrained parametersL←0**for** number of epochs **do**    Sample $S_{Q}$ and $S_{S}$ from $D_{labeled}$ using the Section 3.1 method    Generate prototypes $C_{S}$ using the averaging Equation (Equation 1)    Calculate *L* for the minibatches using the loss Equation (Equation 3)    θ← Update using gradients of *L*▹ compute backpropagation**end for**

## 4. Real-World Case Study

### 4.1. Data Description

CNC milling machines are widely used in a variety of machining industries, commonly known for their precision and high production speed. The dataset in consideration offers a great insight into the complexity and challenges of the CNC machine monitoring use case as it closely represents a real-world industrial case inside a production plant. This work used a publicly available dataset [36] comprising sensor data recorded with the help of a triaxial accelerometer mounted on top of the machining parts of the CNC machine. The data collection was stretched over four different periods of five months each starting from February 2019 to February 2021. Such collection procedures help to tackle the challenges of data drift and the generalization of data-driven approaches. The training and the test dataset host both the normal and abnormal vibration data samples caused by the tool misalignment. Typical process operations that are being carried out by a machining workpiece greatly vary from drilling to cutting. In the scope of this work, each machine hosted 15 different process operations carried out with different physical tools and under a unique configuration. Each sample was a triaxial (X-, Y-, Z-) acceleration data acquired with a sampling rate of 2 kHz. The data were collected from three different CNC machines (M01, M02, and M03) in contention, each containing 15 different process operations ranging from OP00 to OP14. Each data sample was accompanied with the necessary labeling parameters, such as Label, Machine, and Period.

### 4.2. Data Splitting

This section describes the data splitting used in this work. The data were mainly divided into 2 unique sets: training set and test set. The training dataset contained 172 different samples with 156 OK samples and 16 NOK samples, while the test dataset contained 1702 different samples with 1632 OK samples and 70 NOK samples. This reflected the real-world scenario where we generally have a limited labeled data set (training set) with an imbalanced OK/NOK ratio and a relatively large number of unlabeled data (test set). This is illustrated in Figure 4 where the color “violet” represents the training set samples, whereas “orange” denotes the test set samples. To assess the generalization to unseen data and the robustness of the models to data drift, the data splitting was performed according to three different criteria:Machine-wise: This allowed the evaluation of the scalability of the models across different machines. We had 3 CNC Machines in consideration (M01, M02, and M03). Even though, they generated data samples representing the same tool process operations, they varied due to external conditions. Both the training and the test sets were uniformly distributed across the three machines as shown in Figure 4. With a uniform distribution, the model was not offered any unnecessary bias across a particular machine. M03 was not included in the training and was placed aside for testing.Process-wise: This allowed the evaluation of the generalization of the models across unseen tool processes. In industrial applications, new processes are constantly being added due to technological progress and market demand. The training set only contained 4 different tool operations whereas 11 new tool process operations were introduced in the test set.Period-wise: This allowed the evaluation of the robustness of the models across unseen periods. Worn components and aging cause a drift in the data, which affects the data-driven models. For that purpose, the periods of August 2020 and August 2021 were not included in the training and were placed aside for testing.

## 5. Experiments and Analysis

The following section describes the experiments carried out in the scope of this work. The goal of this work was to investigate different strategies for training feature extractors (FEs) for raw industrial time-series data and evaluate them in terms of robustness and generalization. The training of the FE was conducted in a progressive manner. We started by evaluating the performance of the single-stage prototypical network. Once the best parameters were obtained, we proceeded to a comparison with the proposed two-stage model framework. The trained FE models were evaluated on unseen data samples from the test set. We concluded by comparing the FE model obtained by the proposed method with the handcrafted FE and the end-to-end supervised trained FE using a distribution analysis coupled with a feature space analysis. All the experiments were performed under similar conditions with identical training parameters (learning rate = 8 × 10${}^{-4}$, number of epochs = 4, window size $w_{s}$ = 4096 and optimizer = Adam). *N* was fixed to 2 as we only considered two distinct classes for our experiments {Class 1: OK, Class 2: NOK}. While training, the data samples from OP00 to OP04 were separated into the OK and the NOK class sets. During each episode, we randomly picked a number *K* of data samples from these two classes to create the support and query set using the MDS method. The value of *K* representing the number of shots during each episode was varied to determine its effect on the performance of the model. To test generalizability, the models were evaluated using sample data from all available machining operations (OP00 to OP15). The sample data were then picked following the same way as for the training set. The experiments were conducted three times and averaged over their sum to produce the final results. The PN FE models were thereafter evaluated for 1000 episodes of four epochs. The models were trained on a GPU NVIDIA Tesla K80 and generated in Python (version 3.7.4) using the PyTorch library (version 1.8.1).

### 5.1. Single-Stage Prototypical Network

**Experiment**: The single stage prototypical network proceeded with a vanilla implementation of FSL for process failure on the industrial vibration data. The first phase of the experiments used a PN technique with a randomly initialized encoder $f_{\theta }$ with the architecture presented in Figure 3. This experiment was designed to vary two distinctive parameters: *K*, the number of shots, and dist, the distance metric. First, *K* was varied between 1 and 10 shots and the dist was set to the Euclidean distance. Second, dist was varied between Euclidean, DTW, and cosine and *K* was set to seven. Combined with the MDS sampling technique, we focus on obtaining the best set of prototypical learning parameters for industrial vibration data.

**Results**: Table 1 and Table 2 list all the results from the experiments that are compared using different metrics such as “train loss”, “test loss”, “train Accuracy”, “accuracy” (test set), “F1-score” (test set), “precision” (test set), “recall” (test set). For the *K* -shot analysis, all the models converged with 100% accuracy on the training data. The PN model with one-shot learning had the worst F1-score of 76.70%. This is plausible, especially for machining anomaly detection applications, where we face large variations within a single class and often require more samples on the support set to produce better representations (prototypes) and thus a better generalization. The performance of the models gradually increased with the number of shots as can be seen in Table 1. The convergence of the F1-score was reached by the seven-shot PN model at the 87.3% mark. We also noted that the test loss was reduced to 22.76 with a precision score of 89.3%. Upon further increasing the number of shots to 10, we suffered a minimal deterioration of the training loss that can be explained by the drawback of the averaging function performed on the noisy time-series feature vectors. In fact, averaging a relatively large number of deep TS-type features affects the information richness of the prototype vector at some point.

Table 2 compares the results achieved with different distance metrics. With an F1-score just below 54% and a training accuracy of only 66.3%, the DTW-based PN failed to learn. One assumption why the DTW technique failed can be due to the failure to find the best alignment between the prototype vectors and the query vector due to the cyclic behavior of the data. The Euclidean distance, on the other hand, gave the best results, followed by the cosine distance metric, the former getting an F1-score of 87.6% (2.3% higher). This confirmed the findings from Snell et al. [23] for image classification tasks. However, the cosine-based PN offered a better recall (90.5%) over the Euclidean distance recall (85.5%) meaning that it was more reliable in detecting the faulty processes but returned more false positives. This is usually important for industrial applications where quality checks are crucial and demand to be accurate in detecting anomalies, thus prioritizing detecting faulty parts rather than accurately detecting all the good parts.

### 5.2. Two-Stage Prototypical Network

**Experiment**: Using the Euclidean distance and the *K* equal to seven shots, we evaluated and compared the two-stage proposed FE learning framework with the conventional single-stage learning method. Instead of randomly initializing the FE encoder $f_{\theta }$, a pre-training CNN autoencoder was added as an additional layer on top of the prototypical network as stated in Section 3.2. The AE was trained on the full dataset irrespective of the splitting scenario mentioned earlier. This was justified by the fact that today, thanks to IIoT advancements, a huge quantity of unlabeled sensory data is available in the industry and could be used for unsupervised training. We considered a batch size of 32 windows each spanning over 4096 data points which were trained for 40 epochs with a learning rate of 8·$10^{-4}$.

**Results:** The goal of stage one consisted of pretraining the feature extractor $f_{\theta }$ via a CNN AE network in order to break down the complex architecture of high-dimensional sensor data. The results are shown in Figure 5 where we visualize the feature extracted by the $f_{\theta_{pretrained}}$ and the reconstructed signal using $g_{\phi_{pretrained}}$. The reached training loss value is as low as 0.2.

Upon initializing with the learned weights, the PN as part of the two-stage model shows a clear improvement over the single-stage network. It significantly tops the performance chart by achieving an F1-score of 90.3% and an accuracy of 91.0% as shown in Table 3. The two-stage model also proves to have better generalizability on unseen data samples from new class labels as the test loss is significantly decreased from 22.76 for the standard PN to 6.582. It can be further explained with the confusion matrices of both models in Figure 6. The two-stage confusion matrix shows a similar improvement with the inclusion of the pretrained network. The proposed model reaches a prediction rate among its OK samples with an accuracy of 97.62% compared to the single-stage being only at 88%. However, we see a slight deterioration in the NOK accuracy of 1.06%. This can be explained by the pretrained weights in the two-stage training, as the number of OK samples dominated the full dataset over the NOK samples, with an 816:35 imbalance rate. It created a slight bias on the OK samples. We also note a longer training time of 1298 seconds due to the unsupervised pretraining of the feature extraction. The effect of pretraining on the FE model is illustrated in Figure 7. The 2D feature map was generated using a principal component analysis (PCA) [37]. The feature maps changed over time upon increasing the number of epochs. After training four epochs with 1000 episodes each, a clear distinction in the clusters between OK and NOK samples on the two-stage model can be seen in Figure 7, in contrast to the single-stage model where a large number of false positives are observed (in the PC1>1 range). That confirms the results of the confusion matrices from Figure 6.

### 5.3. Handcrafted vs. Supervised vs. Two-Stage FE

**Experiment**: This section provides a detailed comparison of the proposed two-stage model with the handcrafted features and with a feature extractor trained using the traditional end-to-end supervised method. The handcrafted features were extracted using TSFRESH, a state-of-the-art handcrafted feature learning algorithm for industrial time-series data. The supervised method consists of building a classifier block on the top of the feature extraction block presented in Figure 3 and training the network in a conventional end-to-end manner. The classifier block consists of two sequential fully connected layers and a sigmoid as activation function. For the proposed and handcrafted methods, the experiments consisted of training the classifier NN separately using the features extracted with the two-stage FE and TSFRESH, respectively. All the experiments were performed under similar conditions with identical training parameters (epochs: 8, learning rate: 8·10${}^{-4}$, batch size: 32, optimizer: Adam, loss function: binary cross-entropy). The end-to-end supervised method uses the state-of-the-art weight-balancing factor.

**Results:**Table 4 and Figure 8 lay out the results of each of the three methods. The supervised learning delivers the lowest performance among the other methods. Therefore, the F1-score of predicting the correct class only stands at 5.6%. This can be explained by the fact that conventional supervised training requires a huge quantity of labeled data and fails to learn using limited quantity of data. Table 4 shows that features extracted using the two-stage FE outperform the handcrafted FE method with an accuracy of 98.9% (vs. 86.6%) and an F1-score of 88.4% (vs. 84.8%). This performance further highlights the high precision of the proposed method (99.55%) with the classification of the OK class that is shown in the confusion matrix in Figure 8. This confirms the efficiency of the unsupervised pretraining phase where the model learns reliably the dynamic representations of the vibration data and turns more robust against data drift caused by time and wear of the machining components. This can be seen in Figure 9, where the drift across machines and across timeframes is visualized.

The features extracted from the OK class of the exact same process operations using the handcrafted method vary from one machine to another and also over time (when considering the same machine). In contrast, Figure 9 shows the robustness of the two-stage FE, where the OK class data points have the same distribution, with no drift for the across-machine and across-time analysis. We note also that this holds true for the process operations not seen during training (OP06, OP07, and OP12 in Figure 8, as well as for the timeframe (Feb_2020) and the machine (M3) not included in the training set. This result is supported by Table 5, which presents the quantitative analysis of the drift between machines and over time based on the handcrafted features and the deep features extracted by the proposed method. The drift between the *U* and *V* domains was measured using the Wasserstein distance. The two-stage FE shows excellent robustness to drift across the seen and unseen domains within the OK class. We also see a larger distance between the OK and NOK classes, which is consistent with the results from Figure 8.

On the other hand, the handcrafted FE provides less robustness as the distance between the OK domains is not consistent and in some cases, even higher than the distance between the OK and NOK domains. In fact, the OK–NOK Wasserstein distance is equal to 35,869, which is more or less equal to the distances: (M2, M3), (M1, M2), (August 2019, February 2019), and (February 2020, February 2019). A further analysis of Figure 8 reveals the superiority of the proposed two-stage method in OK/NOK separation in the feature space generated by the first two principal components. The two-stage method in Figure 8 shows a clear separation of the normal and abnormal classes compared to the handcrafted and supervised FE methods. It is also important to note that the handcrafted FE has slightly better NOK accuracy, which can be seen in the confusion matrices, with 82.56% compared to 77.47% (two-stage FE). However, the major drawback of the handcrafted technique is the high extraction time (2.2502 s/window) compared to the deep feature learning techniques (0.0054 s/window). This is an important feature for industrial applications that require real-time execution when dealing with real-world use cases.

## 6. Conclusions

In the field of machine condition monitoring, industrial time-series data face major challenges, such as class imbalance, data drift, and most importantly, the lack of pretrained feature extractors. To overcome these challenges, we proposed an efficient two-stage feature learning approach. The proposed technique bridged the gap between unsupervised learning and few-shot learning, which makes it suitable for the industrial scenario where a large quantity of sensory data is available with a limited number of labels. Intuitively adding an autoencoder to a prototype network has proven to be effective. Through a rigorous experimentation and analysis process, we showed that initializing the network with pretrained weights enabled the FE network to upgrade its learning performance. The two-stage learning method produced a feature extractor with higher generalization capabilities compared to the traditional prototypical network, achieving an F1-score of 90.3% with very limited samples. However, it had the disadvantage of a longer training time and a slight decrease in the recall score, while significantly improving the precision score. The research experiments conducted with the traditional prototypical network showed that Euclidean and cosine distance performed best on noisy industrial data, with the Euclidean distance being the best choice in terms of accuracy and the cosine distance in terms of recall. This makes the cosine a better choice for critical quality-testing applications. Finally, the proposed method slightly outperformed the traditional handcrafted feature extractor with an improvement of 4% in the F1-score. Although handcrafted features have the potential to match the performance of the proposed two-stage learning method in terms of classification performance, they have a disadvantage in terms of computation time and robustness to drift. However, this opens the door for future research on hybrid solutions combining handcrafted and deep features. Indeed, extracting handcrafted values from deep features would reduce computation time since it creates a compression of the raw data with the most informative patterns.

## Figures and Tables

**Figure 1 sensors-22-04813-f001:**
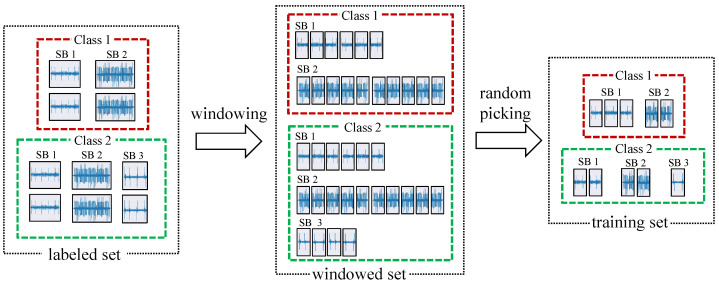
Mixture-based data selection method used in a 2-way and 5-shot FSL for single-axis vibration signals (*C* = 1). SB: subclass.

**Figure 2 sensors-22-04813-f002:**
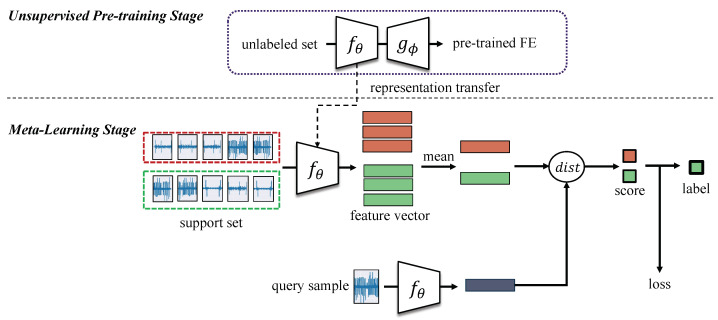
Two-stage few-shot feature learning framework. The OK and NOK classes are shown in green and red respectively.

**Figure 3 sensors-22-04813-f003:**
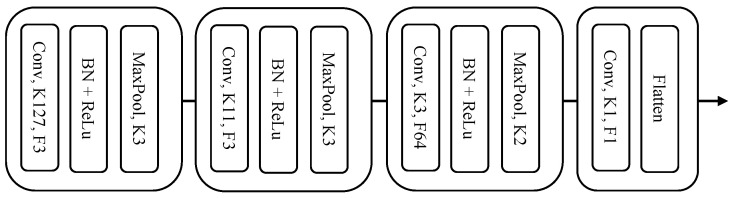
Architecture of the deep feature extractor $f_{\theta }$. MaxPool: maxpooling layer; Conv: 1D convolutional layer; BN: batch-normalization; K: kernel size; F: filter channels.

**Figure 4 sensors-22-04813-f004:**
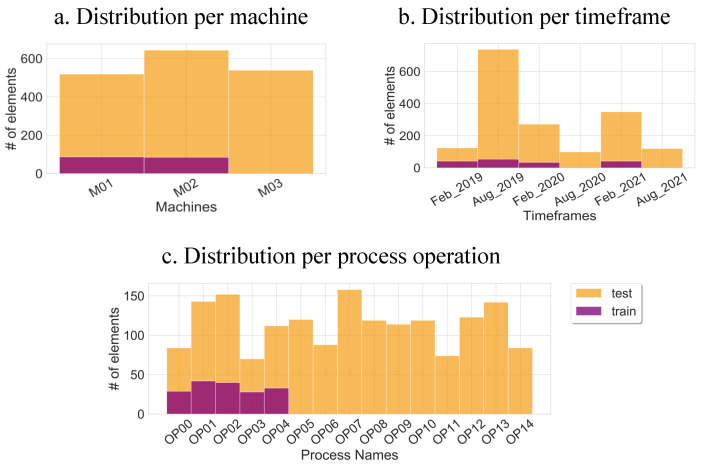
Distribution of the training and test datasets per machine, timeframe, and machining process. To reflect the challenges of industrial practice, a limited amount of data is included in the training set (violet) and unseen data from different machines, time periods and process operations is included in the test set (orange).

**Figure 5 sensors-22-04813-f005:**
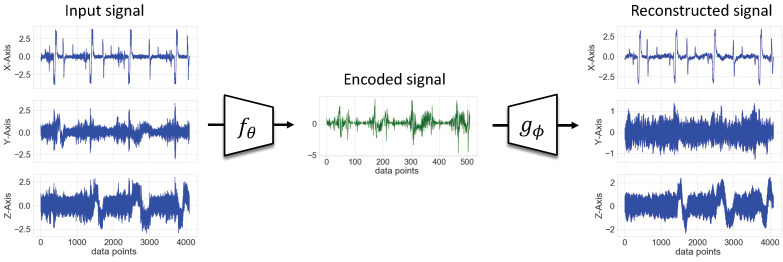
Performance of the resulting AE model at the end of stage one.

**Figure 6 sensors-22-04813-f006:**
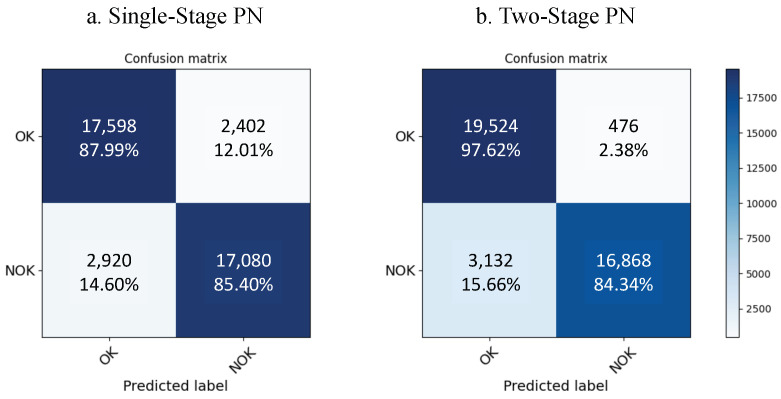
Comparison of the confusion matrices of the single-stage and two-stage learning methods.

**Figure 7 sensors-22-04813-f007:**
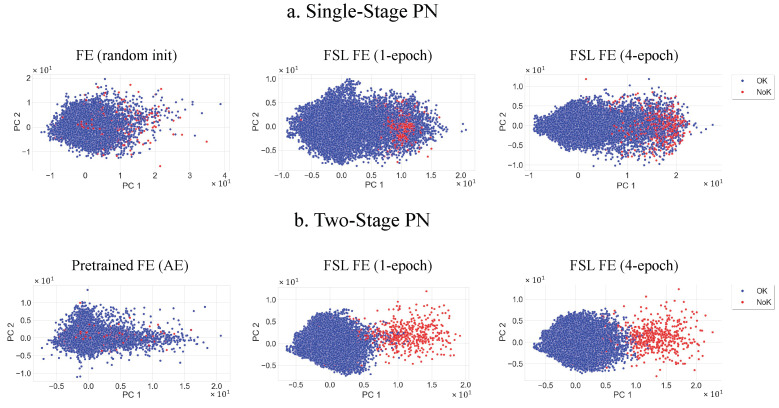
Comparison of the PCA feature spaces of the single-stage and two-stage PN on the pretraining, 1-epoch, and 4-epoch levels. The encoder of the single-stage PN was not pretrained and its parameters were therefore initialized randomly.

**Figure 8 sensors-22-04813-f008:**
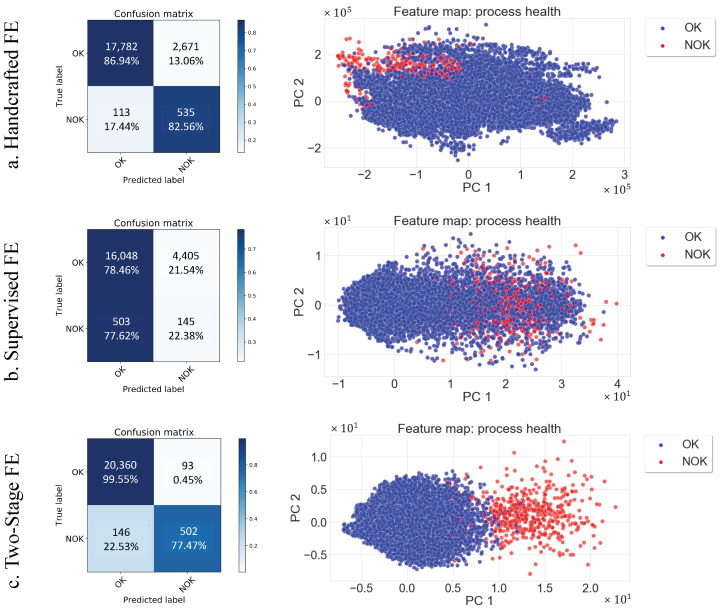
Comparison of the handcrafted (**a**), supervised (**b**), and the proposed method (**c**) trained feature extractors: the top row shows the confusion matrices obtained by the MLP classification, and the bottom row shows the 2D visualization of the FEs’ feature spaces.

**Figure 9 sensors-22-04813-f009:**
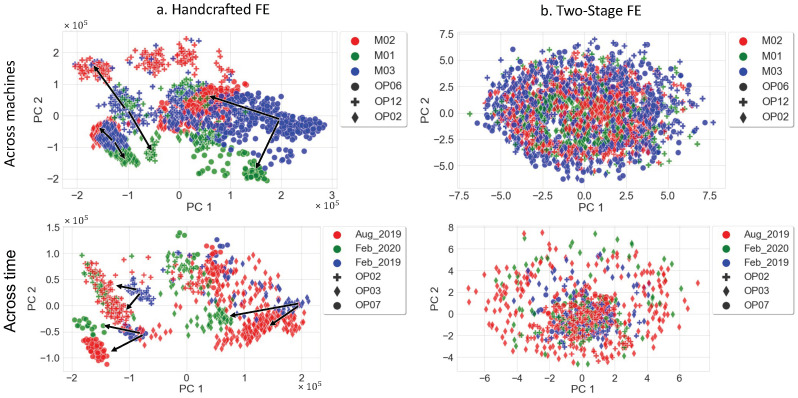
Evaluation of the data drift across machines and time of the handcrafted FE and the proposed method FE within the OK class.

**Table 1 sensors-22-04813-t001:** Results of the *K*-shots experimentation.

Model	Train Loss	Test Loss	Accuracy	F1-Score	Precision	Recall
1-shot	0.1340	87.31	0.765	0.767	0.759	0.773
3-shot	0.0304	36.95	0.848	0.847	0.856	0.835
5-shot	0.0053	29.33	0.860	0.859	0.869	0.848
7-shot	0.0048	22.76	0.876	0.873	0.893	0.855
10-shot	0.0079	21.73	0.882	0.878	0.906	0.853

**Table 2 sensors-22-04813-t002:** Results of the distance measures experimentation.

Distance	Train Accuracy	Accuracy	F1-Score	Precision	Recall
Euclidean	0.999	0.876	0.874	0.894	0.855
DTW	0.737	0.663	0.535	0.865	0.387
Cosine	1.000	0.842	0.851	0.803	0.905

**Table 3 sensors-22-04813-t003:** Evaluation of the proposed method against the single-stage method. The training was run on a GPU NVIDIA Tesla K80.

Model	Test Loss	Accuracy	F1-Score	Precision	Recall	Training Time (s)
Single-Stage	22.76	0.876	0.873	0.893	0.855	295.91
Two-Stage	6.582	0.910	0.903	0.973	0.843	1295.68

**Table 4 sensors-22-04813-t004:** Evaluation of the proposed method against handcrafted and supervised trained feature extractors. The extraction was performed on a CPU Intel Core i7 9850H.

Feature Extractor	Accuracy	F1-Score	Precision	Recall	Training Time per Window (s)
Handcrafted	0.868	0.845	0.862	0.848	2.2502
Supervised	0.767	0.056	0.200	0.032	0.0054
Two-Stage	0.989	0.884	0.905	0.885	0.0054

**Table 5 sensors-22-04813-t005:** Quantitative drift analysis of the handcrafted and two-stage trained feature extractors across time and machines using the Wasserstein distance between the different domains (*U* and *V*).

(a) Handcrafted FE							
	Within only OK class						
**U=**	**Aug_2019**	**Feb_2020**	**Aug_2019**	**M1**	**M1**	**M2**	**OK**
**V=**	**Feb_2019**	**Feb_2019**	**Feb_2020**	**M2**	**M3**	**M3**	**NOK**
	37,680.36	36,229.76	6435.13	37,919.52	14,147.56	37,602.05	35,868.79
(**b**) Two-Stage FE							
	Within only OK class						
**U=**	**Aug_2019**	**Feb_2020**	**Aug_2019**	**M1**	**M1**	**M2**	**OK**
**V=**	**Feb_2019**	**Feb_2019**	**Feb_2020**	**M2**	**M3**	**M3**	**NOK**
	0.484	0.529	0.146	0.337	0.513	1.001	6.22

## Data Availability

Not applicable.

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
