# Peer review of "Efficient Feature Learning Approach for Raw Industrial Vibration Data Using Two-Stage Learning Framework"

_sensors, 2022, doi:10.3390/s22134813_

Round 1
Reviewer 1 Report
1. There are some spelling mistakes in the paper, such as ‘incure’ in the second paragraph of the Introduction and ‘sub-classe’ in the title of Figure 1.
2. What is the meaning of ‘in 3.’ in the first paragraph of section 5.1 of the manuscript?
3. The author should identify channel C in Figure 1. For the convenience of readers, please use the dataset case to describe Mixture-based data selection method clearly in the second paragraph of section 3.1.
4. The meaning of key parameter N should be consistent through the manuscript. However, N represents the number of classes in section 2.1 but the number of shots in section 5.1.
5. Each experiment lacks specific information about the production of data sets. Because it is meta learning, the information of ‘n-way and k-shot’ is very important. Therefore, for each experiment, it is necessary to quantify these information in the support set , such as the number of N and K, how a single sample is sampled, the total number of samples, which OP it comes from, how the data in each OP is used, and the same information in the corresponding test set. The basic and detailed information related to the experiment is logically organized together, which will make it easier for readers to understand the experimental process.
6. The authors claimed that they handle the challenges, such as class imbalance and data drift, but the reasons that the proposed method can address such challenges are not clearly and theoretically described in ‘method description’ section. The authors need to analyze how the method deal with such challenges.
Reviewer 2 Report
Dear Authors,
In the reviewed paper, the Authors presented an effective two-stage feature learning approach to few-shot learning method. Already at the beginning of reading the proposal, you can say that such a split makes sense (shortening the classification time, i.e. also reducing costs). The proposed approach has been verified on data obtained in the actual process of CNC machining. The first and second stage algorithm cannot be faulted. They are exhaustively presented in the following chapters, i.e. descriptions of the smart data sampling technique for the noisy time series and the proposed approach, presentation of the Bosch machining dataset, a proposal of a real scenario mapped in the dataset spliting stages, and finally a description of the experiment performed, the results of which have been thoroughly analyzed and compared with different types of feature extractors. However, the authors did not miss any oversights. Namely, the review of the literature was done very superficially, as no item on two-stage feature selection was cited using a variety of techniques [1 to 14]. Especially those [1 to 5] on the few-shot learning method. In justification of the Authors, it can be mentioned that these papers describe other uses than the peer reviewed paper. Recommendation: Please use Mahalanobis distance in your future work.
Best regards, Reviewer.
1) Debasmit Das, C. S. George Lee: A Two-Stage Approach to Few-Shot Learning for Image Recognition. IEEE TRANSACTIONS ON IMAGE PROCESSING, VOL. 29, 2020
2) W. ZI, L. S. GHORAIE, S. PRINCE: Tutorial # 2: few-shot learning and meta-learning I Oct. 8, 2019. https://www.borealisai.com
3) Arman Afrasiyabi, Jean-Franc¸ois Lalonde, Christian Gagne´: Mixture-based Feature Space Learning for Few-shot Image Classification.
https://openaccess.thecvf.com/content/ICCV2021/papers/Afrasiyabi_Mixture-Based_ ...
4) Runhao Jiang, Jie Zhang, Rui Yan, Huajin Tang: Few-Shot Learning in Spiking Neural Networks by Multi-Timescale Optimization. Neural Computation 2021) 33 (9): 2439–2472
5) J. Ma, H. Xie, G. Han, et.al .: Partner-Assisted Learning for Few-Shot Image Classification. https://openaccess.thecvf.com/content/ICCV2021/papers/Ma_Partner-Assisted_ Learning_ ....
6) Zhuang Li, Xincheng Tian, ​​Xin Liu, Yan Liu and Xiaorui Shi: Two-Stage Industrial Defect Detection Framework Based on Improved-YOLOv5 and Optimized-Inception-ResnetV2 Models. Appl. Sci. 2022, 12, 834
7) Z. Shen, Z. Liu, Jie Qin, et.al .: Partial Is Better Than All: Revisiting Fine-tuning Strategy for Few-shot Learning. www.aaai.org/AAAI21Papers/AAAI-1041.ShenZ.pdf
8) Hongling Xu, Ruizhe Ma, Li Yan, Zongmin Ma: Two-stage prediction of machinery fault trend based on deep learning for time series analysis. Digital Signal Processing Volume 117, 2021, 103150
9) Shangjie Ren; Kai Sun; Chao Tan; Feng Dong: A Two-Stage Deep Learning Method for Robust Shape Reconstruction With Electrical Impedance Tomography. IEEE Transactions on Instrumentation and Measurement (Volume: 69, Issue: 7, July 2020, 4887 - 4897
10) Shahabodin Afrasiabi; Mousa Afrasiabi; Benyamin Parang; Mohammad Mohammadi: Two-Stage Deep Learning-based Wind Turbine Condition Monitoring Using SCADA Data. 2020 IEEE International Conference on Power Electronics, Drives and Energy Systems (PEDES)
11) Athina Tsanousa, Evangelos Bektsis, Constantine Kyriakopoulos; et al: A Review of Multisensor Data Fusion Solutions in Smart Manufacturing: Systems and Trends. Sensors 2022, 22, 1734.
12) X. Zhang, Q. Zhang, M. Chen, Y. Sun, X. Qin, and H. Li, A two-stage feature selection and intelligent fault diagnosis method for rotating machinery using hybrid filter and wrapper method, Neurocomputing, vol. 275, pp. 2426–2439, 2018
13) K. Yu, T. R. Lin, and J. Tan, “A bearing fault and severity diagnostic technique using adaptive deep belief networks and Dempster-Shafer theory, Structural Health Monitoring, vol. 19, no. 1, pp. 240–261, 2020.
14) C. Yiakopoulos, K. Gryllias, and I. Antoniadis, “Rolling element bearing fault detection in industrial environments based on a k-means clustering approach,” Expert Systems with Applications, vol. 38, no. 3, pp. 2888 - 2911, 2011.
